# Interstitial Lung Disease and Anti-Myeloperoxidase Antibodies: Not a Simple Association

**DOI:** 10.3390/jcm10122548

**Published:** 2021-06-09

**Authors:** Marco Sebastiani, Fabrizio Luppi, Gianluca Sambataro, Diego Castillo Villegas, Stefania Cerri, Paola Tomietto, Giulia Cassone, Marialuisa Bocchino, Belen Atienza-Mateo, Paolo Cameli, Patricia Moya Alvarado, Paola Faverio, Elena Bargagli, Carlo Vancheri, Miguel A. Gonzalez-Gay, Enrico Clini, Carlo Salvarani, Andreina Manfredi

**Affiliations:** 1Rheumatology Unit, Azienda Ospedaliera Policlinico di Modena, University of Modena and Reggio Emilia, 41121 Modena, Italy; giulia.cassone@unimore.it (G.C.); carlo.salvarani@unimore.it (C.S.); Andreina.Manfredi@gmail.com (A.M.); 2Department of Medicine and Surgery, University of Milano Bicocca, Respiratory Unit, San Gerardo Hospital, ASST di Monza, 20900 Monza, Italy; fabrizio.luppi@unimib.it (F.L.); paola.faverio@unimib.it (P.F.); 3Regional Referral Centre for Rare Lung Diseases, A.O.U. “Policlinico-Vittorio Emanuele”, Department of Clinical and Experimental Medicine, University of Catania, 95123 Catania, Italy; dottorsambataro@gmail.com (G.S.); vancheri@unict.it (C.V.); 4Respiratory Department, Hospital de la Santa Creu i Sant Pau, 08041 Barcelona, Spain; DCastillo@santpau.cat (D.C.V.); pmoyaa@santpau.cat (P.M.A.); 5Respiratory Unit, University of Modena and Reggio Emilia, 41121 Modena, Italy; stefania.cerri@unimore.it (S.C.); enrico.clini@unimore.it (E.C.); 6Rheumatology Unit, University of Trieste, 34127 Trieste, Italy; paola.tomietto@asuits.sanita.fvg.it; 7PhD Program in Clinical and Experimental Medicine, University of Modena and Reggio Emilia, 41121 Modena, Italy; 8Department of Clinical Medicine and Surgery, Federico II University Hospital, 80131 Naples, Italy; marialuisa.bocchino@unina.it; 9Department of Rheumatology, Hospital Universitario Marqués de Valdecilla, IDIVAL, 39011 Santander, Spain; mateoatienzabelen@gmail.com (B.A.-M.); miguelaggay@hotmail.com (M.A.G.-G.); 10Respiratory Diseases Unit, Department of Medical Sciences, Surgery and Neurosciences, Siena University, 53100 Siena, Italy; paolocameli88@gmail.com (P.C.); bargagli2@gmail.com (E.B.); 11Cardiovascular Pathophysiology and Genomics Research Unit, School of Physiology, Faculty of Health Sciences, University of the Witwatersrand, Johannesburg 2094, South Africa; 12Rheumatology Unit, Irccs Arcispedale Santa Maria Nuova, Azienda Unità Sanitaria Locale-IRCCS di Reggio Emilia, 42124 Reggio Emilia, Italy

**Keywords:** interstitial pneumonia, idiopathic pulmonary fibrosis, vasculitis, rheumatic diseases, anti-myeloperoxidase antibodies

## Abstract

Anti-neutrophil cytoplasmic antibodies (ANCA), mainly anti-myeloperoxidase (MPO) antibodies, have been frequently identified in patients with idiopathic pulmonary fibrosis (IPF). However, their role remains unclear, and only 7–23% of these patients develops clinically overt vasculitis. We aimed to investigate the clinical, serological, and radiological features and prognosis of anti-MPO-positive interstitial lung disease (ILD) patients. Fifty-eight consecutive patients firstly referred for idiopathic interstitial pneumonia and showing serological positivity of anti-MPO antibodies were retrospectively enrolled. For each patient, clinical data, lung function testing, chest high-resolution computed tomography (HRCT) pattern, and survival were recorded. Thirteen patients developed a rheumatic disease during a median follow-up of 39 months. Usual interstitial pneumonia (UIP) was the most frequent ILD pattern, significantly influencing the patients’ survival. In fact, while the 52-week survival of the overall population was 71.4 ± 7.5%, significantly higher than IPF, survivals of anti-MPO patients with UIP pattern and IPF were similar. Forced vital capacity and diffusion lung capacity for CO significantly declined in 37.7 and 41.5% of cases, respectively, while disease progression at chest HRCT was observed in 45.2%. A careful clinical history and evaluation should always be performed in ILD patients with anti-MPO antibodies to quickly identify patients who are developing a systemic rheumatic disease.

## 1. Introduction

Anti-neutrophil cytoplasmic antibodies (ANCA) are autoantibodies specific for antigens located in the cytoplasmic granules of neutrophils and lysosomes of monocytes [1]. Myeloperoxidase (MPO) and proteinase 3 (PR3) are the two main autoantigen targets. ANCA are generally detected in multisystemic diseases, namely ANCA-associated vasculitides (AAV), characterized by necrotizing vasculitis typically involving small vessels [2]. After the first description in 1990, the association between interstitial lung disease (ILD) and AAV was confirmed in 1994 in a Japanese study that reported a 43% prevalence of ILD in MPO-ANCA-positive patients with collagen vascular disease and or glomerulonephritis [3].

On the other side, an increasing number of authors reported the presence of ANCA in patients with idiopathic pulmonary fibrosis (IPF), with a prevalence ranging from 1 to 35%, mainly in the Japanese population [4,5,6]. Anti-MPO antibodies are the most frequent ANCA subtype described in these patients. Moreover, ANCA conversion may occur in up to 11% of patients during the IPF course [7]. The role of anti-MPO antibodies in patients with idiopathic interstitial pneumonia (IIPs) remains unclear. The occurrence of vasculitis, mainly microscopic polyangiitis (MPA), has been described in a minority of cases. Indeed, only 7–23% of ANCA-positive subjects subsequently developed a clinically overt vasculitis during the follow-up, mainly limited to patients with anti-MPO antibodies [8,9].

Conflicting data have been reported about the prognostic role of ANCA antibodies in IPF patients. Some authors reported no difference in survival, while others observed a poorer prognosis in ANCA-positive IPF patients, regardless of the appearance of vasculitis [5,7,8,10,11].

This multicenter, retrospective study aimed to investigate the clinical, serological, and radiological features, as well as the prognosis of anti-MPO-positive ILD patients.

## 2. Materials and Methods

### 2.1. Study Population and Data Collection

We retrospectively collected the clinical charts of all consecutive anti-MPO-positive patients from 6 Italian and Spanish centers with a special interest in ILD, firstly referred for IIP from 1 January 2015 to 31 December 2019. All patients showed a positivity for anti-MPO antibodies at the time of ILD diagnosis. The search for anti-MPO antibodies was performed by combining indirect immunofluorescence and ELISA using tests with similar specificity and sensitivity in the different centers. Patients with a previous diagnosis of connective tissue disease (CTD) or systemic vasculitis were excluded from the study. The study was conducted according international ethics standard and was approved by the local ethical committee; patients gave their informed consent to be enrolled.

Data about clinical manifestation suggestive of vasculitis or CTD, lung function testing, autoimmunity data, and CT imaging pattern were recorded for each patient at the time of the first and of the last available visit.

Based on previous reports on classification of patterns of pulmonary involvement in chest high-resolution tomography (HRCT), patient chest HRCT findings were categorized into usual interstitial pneumonia (UIP), nonspecific interstitial pneumonia (NSIP), organizing pneumonia (OP), or other patterns that can also be observed in combination [12,13,14]. All HRCT images were locally reviewed by an expert thoracic radiologist.

Lung function and CT imaging variations were recorded for the observation period available for each patient. An absolute decline >10% of the forced vital capacity (FVC) or >15% of the lung diffusion capacity of CO from baseline were considered clinically significant [12]. Improvement, worsening, or stability of chest HRCT was evaluated according to the percentage of involvement of the lung parenchyma [15]. Main disease-specific treatments, including immune-suppressants and anti-fibrotic agents, were recorded as well. The end of the follow-up was set at the last available visit or when progression to AAV or death occurred. A patient was defined as lost to follow-up when he/she had no follow-up visit or when the follow-up duration was lower than 12 months. At the end of follow-up, all diagnoses were discussed in a multidisciplinary discussion involving rheumatologist and pulmonologist.

Finally, the survival of MPO-positive-ILD patients was compared with that of a reference cohort of 116 Italian age-matched patients with anti-MPO-negative IPF referred to the University of Modena.

### 2.2. Statistical Analysis

Results were expressed as median and interquartile range (IQR) for numerical data, while frequency was reported as number and percentage.

Categorical variables were analyzed by Chi-square test or Fisher’s exact test and differences between the means were determined using the Mann–Whitney test for unpaired samples. *p* values < 0.05 were considered statistically significant. Statistical analyses were performed using the SPSS statistical software, version 17.0 (SPSS Inc., Chicago, IL, USA) [16].

## 3. Results

Fifty-eight patients were enrolled in the study (F/M 27/31; median age 69 years, IQR 9) and observed for a median follow-up of 39 months (IQR 51). The baseline characteristics of the study population are reported in Table 1.

At the first observation, only nonspecific symptoms and signs of vasculitis or CTD were recorded, namely sicca syndrome and Raynaud’s phenomenon, with no evidence of any systemic involvement. Three patients (3/58 = 5%) satisfied the criteria for a diagnosis of interstitial pneumonia with autoimmune features (IPAF) [17,18]. Low-titer antinuclear antibodies (ANA) were detected in 19 patients (32.7%), without significant extractable nuclear antigen (ENA) specificities.

The baseline ventilatory pattern was preserved in the majority of cases, with FVC values >70% of those predicted in 45 subjects (77.6%). Conversely, gases exchange was impaired with estimated DLCO values >80% of predicted limited to seven patients (12.1%).

Usual interstitial pneumonia was the most frequent radiologic pattern of ILD at presentation, being observed in more than half of patients (51.7%). An NSIP was recorded in 11 cases (19%), OP in four (6.9%), and other or combined patterns in the remaining 22.4% of patients.

Overall, five patients had only one visit and were classified as lost to follow-up; the remaining patients (91%) had a median follow-up period of 39 months (IQR 50). Among these, 22 (41.5%) received at least one immune-suppressive drug, mainly azathioprine or mycophenolate mofetil, whereas 13 (24.5%) were treated with anti-fibrotic drugs, as detailed in Table 2. Finally, 34 patients (64.1%) received chronic systemic corticosteroids in five cases in association with anti-fibrotic drugs. Treatment with anti-fibrotic was exclusively limited to patients with a chest CT UIP pattern, while immune-suppressants were used regardless of imaging features.

Lung function slightly deteriorated over time in the whole study population. Median FVC declined from 84 to 80% of the predicted value (*p* = 0.031), while predicted DLCO passed from 55 to 49% (*p* < 0.001).

In more details, a decline in FVC > 10% was observed in 20 out of 53 patients (37.7%), while it improved in eight (15.1%). A significant reduction in DLCO (>15%) occurred in 22 patients (41.5%), while it was ameliorated in five (9.4%). Consistently, disease extension assessed by chest HRCT worsened in 24 patients (45.2%) and improved in nine (16.9%).

After a median follow-up period of 27 months (range 12–111), nine out 53 (17%) patients developed a definite AAV, and an additional four cases (7%) developed another rheumatic disease, namely Sjogren’s syndrome in three cases and rheumatoid arthritis in one (Figure 1).

No differences were observed in baseline features between patients who did or did not develop a rheumatic disease. In particular, sicca syndrome and arthralgias were not predictive of progression to vasculitis or CTD. Only six out of 19 patients (36%) with baseline ANA developed a rheumatic disease (*p* not significant when compared with ANA-negative patients).

Finally, the median survival time for ILD-MPO patients was 192 months. The overall cohort of anti-MPO-positive ILD patients had a better 52-week survival in comparison with 116 age-matched reference IPF cases (71.4 ± 7.5% vs. 52.4 ± 5.9% for anti-MPO-positive ILD patients and IPF, respectively; *p* = 0.045). Conversely, no survival differences with IPF were observed when looking at MPO-positive ILD patients with a chest CT pattern of UIP (62.4 ± 12.3%; Figure 2).

## 4. Discussion

In 1999, Becker-Merok et al. described an IPF patient developing glomerulonephritis and necrotizing vasculitis, as well as positivity for p-ANCA antibodies [19].

Afterward, many authors suggested an association between IPF and anti-MPO antibodies, regardless of the occurrence of vasculitis, mainly micro-polyangiitis (MPA) [4,5,7,8,10,11].

In IPF patients, a positivity for ANCA antibodies may be observed in about 5–10% of cases [4,5]. In particular, anti-MPO antibodies have been reported in 4–36% of patients with idiopathic interstitial pneumonia, including IPF, while the occurrence of MPA has been described in a range from 1.7 to 25.7% of patients [20,21]. This group of patients might be called “pulmonary limited vasculitis” and considered a phenotypic variant of MPA, as reported by Katsumata et al. [22].

ILD appearance follows the diagnosis of MPA only in a minority of cases. At the same time, in more than 80%, it is diagnosed concurrently or before vasculitis [3,10,23,24,25,26], with an interval of time between diagnoses of ILD and vasculitis ranging from a few months to many years. In our population, 13 patients developed a rheumatic disease. Of interest, an AAV was only observed in nine patients, MPA in all cases, whereas the other three developed Sjogren’s syndrome and the latter rheumatoid arthritis.

Interestingly, in our population, no serological or clinical features were associated with the progression to a rheumatic disease. Few studies described the possible association of anti-MPO antibodies with RA or SS. Anti-MPO antibodies have been described to occur in a low number of RA patients, without any change in clinical and radiologic features of the disease [27], while association with SS is very rare and mainly limited to patients with concurrent AAV [28,29].

Interestingly, in our population, no serological or clinical features were associated with the progression to a rheumatic disease. This is particularly true for patients evolving in MPA. In contrast, in patients developing RA and SS, ANA were positive at baseline, and some clinical manifestations, such as arthralgia and sicca syndrome, were recorded. On the other side, only a few patients with ANA positivity developed a rheumatic disease in the subsequent years. Anyway, a careful clinical and anamnestic evaluation should always be performed to identify patients at risk for evolution in a systemic rheumatic disease. Moreover, a systematic search for complement fractions, renal function, and inflammation indexes, such as C-reactive protein, should be assessed in all patients with anti-MPO antibodies.

For the absence of evidence-based treatment, the therapy proposed to our patients was very heterogeneous, reflecting the single center’s experience. Radiological patterns of ILD seem to be the main driver of therapeutic choice; in fact, antifibrotic drugs, namely nintedanib and pirfenidone, have been prescribed in 13 patients, all with UIP pattern. On the other side, administration of immunosuppressants did not appear related to the lung disease features, because this treatment was prescribed regardless of radiological pattern. Moreover, the treatment did not seem to affect lung disease progression. Among nine patients with an improvement of HRCT features, only three had been treated (with azathioprine, mycophenolate mofetil, and pirfenidone, respectively), while an immunosuppressive (10 patients) or antifibrotic (eight patients) therapy was prescribed in patients with disease worsening. Therefore, we can suppose that lung deterioration and severity, more than the ILD pattern, could represent the driver for starting treatment. Of interest, patients initially treated with an immunosuppressant changed their immunosuppressive drug when deterioration of lung function was recorded, but no patients were shifted from immunosuppressive to antifibrotic therapy or vice versa.

While the presence of ILD with a UIP pattern significantly reduces the survival of patients with ANCA-associated vasculitis [30], conflicting data have been reported about the possible influence of ANCA on the survival of IPF patients. In fact, many authors described similar survival rates for patients positive or negative for ANCA [7,8,10,11,22,26], while others suggested a worse prognosis for ANCA-positive patients, mainly anti-PR3 antibodies [19].

In a retrospective study, 12 patients with MPO-ANCA/UIP were compared with 108 IPF/UIP patients [11]. Despite the limitation of the study, the authors did not observe differences between the two populations, including survival and frequency of acute exacerbations [11]. Our results confirm that the radiologic ILD pattern is the best predictor of survival; in particular, in patients with UIP pattern, the presence of anti-MPO did not modify survival when compared with patients with IPF.

Recently, the presence of MPA has been reported as a cause of worst prognosis in patients with interstitial pneumonia and anti-MPO antibodies [31]. The exclusion of patients with a previous diagnosis of vasculitis and the low number of patients that developed a MPA does not allow us to evaluate this point.

The main limitation of our study is its retrospective design; furthermore, five patients were lost at follow-up reducing the number of the evaluated patients. However, this is one of the largest studies investigating this specific population. Moreover, our control group included only Italian IPF patients; although no data were reported in literature about possible differences in clinical history or survival of Spanish and Italian IPF patients, we cannot completely exclude a possible bias. Our results confirm the need to always evaluate patients with idiopathic interstitial pneumonia for ANCA, not only at the beginning of the study but also during follow-up. On the other hand, the heterogeneity in the treatment choices highlights the necessity of a consensus for the management of this specific population. Finally, at baseline, three patients satisfied research criteria for IPAF, but anti-MPO antibodies are not currently included in the serologic domain of classification IPAF criteria [17,18]. The possible evolution in a definite rheumatic disease and the unresolved questions about the treatment of these patients should be taken into account in a future revision of the IPAF criteria, evaluating the inclusion of anti-MPO antibodies.

## 5. Conclusions

Anti-MPO-ILD patients are at risk of progression in a definite rheumatic disease, mainly ANCA-associated vasculitis, in a variable range of time. Therefore, their follow-up should always include a careful clinical–laboratorial evaluation exploring rheumatic disorders. Only prospective longitudinal studies including a large number of patients may allow us to identify possible associated factors to the evolution towards vasculitis or other inflammatory rheumatic diseases and the prognostic role of anti-MPO antibodies in patients with idiopathic interstitial pneumonia.

## Figures and Tables

**Figure 1 jcm-10-02548-f001:**
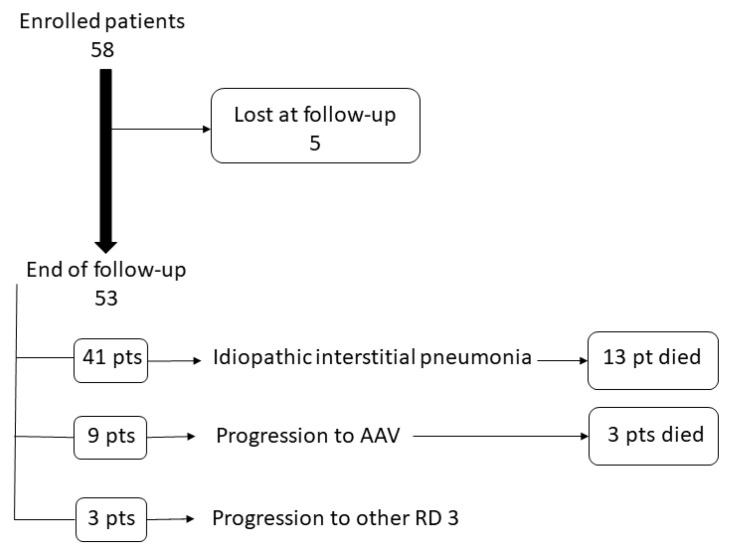
Clinical evolution of patients enrolled in the study.

**Figure 2 jcm-10-02548-f002:**
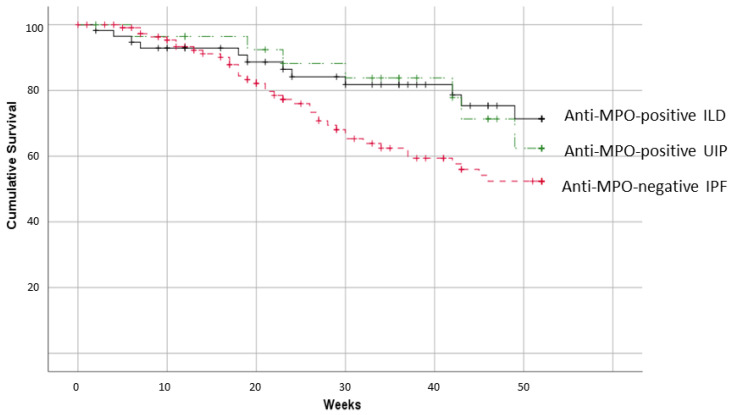
Fifty-two-week survival of patients with anti-MPO, anti-MPO-UIP, and IPF.

**Table 1 jcm-10-02548-t001:** Baseline features of 58 anti-MPO-ILD patients.

	*n*	%
Sex M/F	27/31	46.6/53.4
Smoking history	23	39.7
ILD pattern		
UIP	30	51.7
NSIP	11	19.0
OP	4	6.9
Mixed pattern	13	22.4
	Median	IQR
Age at ILD diagnosis	69	9
Follow-up duration (months)	39.0	51
FVC baseline (%)	84	28.25
FVC end follow-up (%)	80	29
DLCO baseline (%)	55	27.0
DLCO end follow-up (%)	49	36.0

Anti-MPO: anti-myeloperoxidase antibodies; ILD: interstitial lung disease; M: males; F: females; UIP: usual interstitial pneumonia; NSIP: nonspecific interstitial pneumonia; OP: organizing pneumonia; FVC: forced vital capacity; DLCO: diffusion lung capacity for CO; IQR: interquartile range.

**Table 2 jcm-10-02548-t002:** Proposed treatments in 58 anti-MPO-ILD patients.

	*n*	%
Azathioprine	11	20.7
Cyclophosphamide	4	7.5
Mycophenolate	8	15.1
Rituximab	3	5.7
No immunosuppressors	31	58.5
1 immunosuppressor	19	35.8
2 immunosuppressors	2	3.8
3 immunosuppressors	1	1.9
Pirfenidone	8	15.1
Nintedanib	5	9.4
Steroids	34	64.1
O_2_ supplementation	18	34

## Data Availability

Data are available if requested.

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
