# Peer review of "Interstitial Lung Disease and Anti-Myeloperoxidase Antibodies: Not a Simple Association"

_jcm, 2021, doi:10.3390/jcm10122548_

Round 1

Reviewer 1 Report

Dear Authors,

I read with great interest this Your manuscript, submitted for publication.

I have few comments. I hope that they can be useful. 

  1. The survival of MPO-positive-ILD patients was compared wuth that of a reference cohort of Italian patients with anti-MPO-negative IPF referred to the University of Modena. You retrospectively collected data from six Italian and Spanish centers. My comment is: was Your reference cohort representative of all the different regional and national realities ? Is it possible that this is a referral/reference bias in Your study ?
  2. In Your study, 3 patients developed Sjogren's syndrome and 1 rheumatoid arthritis. Please, discuss the relationship between anti-MPO-positive ILD and these rheumatic diseases
  3.  Please, add IPAF after "interstitial pneumonia with autoimmune features"  (please, see line 120)   

Author Response

The survival of MPO-positive-ILD patients was compared wuth that of a reference cohort of Italian patients with anti-MPO-negative IPF referred to the University of Modena. You retrospectively collected data from six Italian and Spanish centers. My comment is: was Your reference cohort representative of all the different regional and national realities ? Is it possible that this is a referral/reference bias in Your study ?

We thank the referee for this interesting comment. Although no differences were reported in literature between clinical history or survival of Spanish and Italian IPF patients, together with the availability of worldwide guidelines on IPF diagnosis and treatment, we include this point among the limitations of the study.

    In Your study, 3 patients developed Sjogren's syndrome and 1 rheumatoid arthritis. Please, discuss the relationship between anti-MPO-positive ILD and these rheumatic diseases

This is a very interesting comment, since association between Sjogren's syndrome, rheumatoid and anti-MPO-antibodies is very rare. We added a comment in the discussion

 Please, add IPAF after "interstitial pneumonia with autoimmune features"  (please, see line 120)  

We have added “IPAF” according to the referee’s suggestion

Reviewer 2 Report

Authors retrospectively analyzed data of a consistent group of ILD patients with anti MPO antibodies collected from six Southern Europe Centers. The aim of the study was to investigate the clinical, serological, and radiological features, as well as the prognosis of anti-MPO-positive ILD patients.  Manuscript is clear and well written; a few changes might improve it.

COMMENTS:

  1. Enrolled patients were anti MPO positive, but did ILD diagnosis and anti MPO detection occur at the same time? If not how long before/after?
  2. A flow chart or a table with all the enrolled patients and patients who developed ADD or died during the study could be useful to summarize data.
  3. Did authors have data regarding anti MPO changes during the follow-up? Did a reduction after steroid or other immunosuppressive therapies occur?
  4. Were levels of anti MPO at recruitment different between ILD patients who developed AAV and those who did not present it?
  5. Did authors only analyze two time points: enrollment= visit 0 and follow-up= visit 1 or more? Please clarify this point in the “Study population and data collection” section.
  6. Authors should add the method(s) used to evaluate anti MPO with sensitivity and specificity and, in case of different methods, if accuracy was similar among them.
  7. In figure 1 authors should add “negative (-) anti MPO IPF,+ anti MPO ILD and + anti MPO UIP “ to make clearer the analyzed groups.
  8. Line 78: please correct according to
  9. Line 227 IPAF please indicate the abbreviation “Interstitial pneumonia with autoimmune features”

Author Response

Authors retrospectively analyzed data of a consistent group of ILD patients with anti MPO antibodies collected from six Southern Europe Centers. The aim of the study was to investigate the clinical, serological, and radiological features, as well as the prognosis of anti-MPO-positive ILD patients.  Manuscript is clear and well written; a few changes might improve it.

COMMENTS:

  1. Enrolled patients were anti MPO positive, but did ILD diagnosis and anti MPO detection occur at the same time? If not how long before/after?

We thank the referee for this comment. Anti-MPO positivity was referred to the time of diagnosis of ILD, as screening for secondary forms of ILD. We added this point in the text

  1. A flow chart or a table with all the enrolled patients and patients who developed AVV or died during the study could be useful to summarize data.

We added a figure

  1. Did authors have data regarding anti MPO changes during the follow-up? Did a reduction after steroid or other immunosuppressive therapies occur?

We thank the referee for this comment. This is an interesting point. Unfortunately, these data are not available. In patients enrolled in our study anti-MPO were measured only with a diagnostic purpose and therefore not detected during the follow-up.

  1. Were levels of anti MPO at recruitment different between ILD patients who developed AAV and those who did not present it?

We thank the Reviewer for raising this important point. Our data do not suggest a correlation between anti-MPO levels and future development of a specific AAV.

  1. Did authors only analyze two time points: enrollment= visit 0 and follow-up= visit 1 or more? Please clarify this point in the “Study population and data collection” section.

We thank the referee for the comment. We clarified this point in the text. Patients were analysed at the time of the first and of the last available visit

  1. Authors should add the method(s) used to evaluate anti MPO with sensitivity and specificity and, in case of different methods, if accuracy was similar among them.

Search for ANCA was made according to the “Revised 2017 international consensus on testing of ANCAs in granulomatosis with polyangiitis and microscopic polyangiitis”.

We combined results obtained by indirect immunofluorescence and ELISA with a specificity and a sensitivity of the tests in the different centers ranging between 96–99% for specificity and 71–88% for sensitivity, respectively, in patients with AAV.

  1. In figure 1 authors should add “negative (-) anti MPO IPF,+ anti MPO ILD and + anti MPO UIP “ to make clearer the analyzed groups.

We modified the figure according to the referee’s suggestion

  1. Line 78: please correct according to

We have corrected the mistake

  1. Line 227 IPAF please indicate the abbreviation “Interstitial pneumonia with autoimmune features”

We have added the abbreviation